# Directly Printed Low-Cost Nanoparticle Sensor for Vibration Measurement during Milling Process

**DOI:** 10.3390/ma13132920

**Published:** 2020-06-29

**Authors:** Soo-Hong Min, Tae Hun Lee, Gil-Yong Lee, Daniel Zontar, Christian Brecher, Sung-Hoon Ahn

**Affiliations:** 1Department of Mechanical Engineering, Seoul National University, Seoul 08826, Korea; msh7799@snu.ac.kr; 2Department of Production Machines, Fraunhofer Institute for Production Technology IPT, Aachen 52074, Germany; daniel.zontar@ipt.fraunhofer.de (D.Z.); christian.brecher@ipt.fraunhofer.de (C.B.); 3Department of Mechanical Engineering, Kumoh National Institute of Technology, Gumi 39177, Korea; gylee@kumoh.ac.kr; 4Institute of Advanced Machines and Design, Seoul National University, Seoul 08826, Korea

**Keywords:** milling, workpiece, direct printing, vibration, sensor

## Abstract

A real-time, accurate, and reliable process monitoring is a basic and crucial enabler of intelligent manufacturing operation and digital twin applications. In this study, we represent a novel vibration measurement method for workpiece during the milling process using a low-cost nanoparticle vibration sensor. We directly printed the vibration sensor based on silver nanoparticles positioned onto a polyimide substrate using an aerodynamically-focused nanomaterials printing system, which is a direct printing technique for inorganic nanomaterials positioned onto a flexible substrate. Since it does not require any post-process such as chemical etching and heat treatment, a highly sensitive vibration sensor composed of a microscale porous structure was fabricated at a cost of several cents each. Furthermore, accurate and reliable vibration data was obtained by simple and direct attachment to a workpiece. In this study, we discussed the performance of vibration measurement of a fabricated sensor in comparison to a commercial vibration sensor. Using frequency and power spectrum analysis of obtained data, we directly measured the vibration of workpiece during the milling process, according to a process parameter. Lastly, we applied a fabricated sensor for the digital twins of turbine blade manufacturing in which vibration greatly affects the quality of the product to predict the process defects in real-time.

## 1. Introduction

Following the trends to Industry 4.0 and a smart manufacturing system, there is a growing demand of digitalised and automated manufacturing process in several high technology fields including the aerospace and automobile industry [1,2,3]. Furthermore, an intelligent manufacturing process to increase productivity is essential, especially for manufacturing companies in high-wage countries that are under pressure with regard to international competition due to the lower production costs in low-wage countries [4]. Inter alia, mechanical machining is a process with one of the greatest potentials with regard to automation in terms of process time and cost in industry. Hence, there have been several efforts to optimise machining process lines automatically, which require monitoring of each process [5,6,7]. In this respect, basic and crucial enablers of smart and intelligent machining operation include real-time, accurate, and reliable process monitoring.

In general, cutting force, machining vibration, tool wear, and surface integrity are the main targets for process monitoring in machining [8,9,10]. Conventionally, dynamometers and accelerometers have been used for direct measurement of cutting forces and mechanical vibrations, respectively [11,12,13]. Moreover, indirect measurement methods including acoustic emission measurement and power consumption measurement have been proposed for machining process monitoring [14,15]. However, these approaches usually require expensive sensors and the data acquisition system make them difficult to use for industrial applications and limits their usability in specific machining environments or conditions.

Recently, small and low-cost sensors for machining process monitoring have been developed to overcome the above technological and industrial limitations including piezo electric sensors fibre Bragg grating sensor and acoustic wave sensor [16,17,18,19]. These achievements had mainly focused on the compact integration with existing equipment or machine to minimise the effect on the process or lowering sensor fabrication cost. However, installation of these sensors to machining equipment for process monitoring requires additional efforts or cost.

In this study, we developed novel vibration sensors with low-cost and directly attachable properties for deeper understanding of the workpiece vibration and error minimisation in milling process monitoring. The vibration sensors were fabricated by aerodynamically-focused nanomaterials (AFN) printing system, which is a direct nanomaterials printing method in a low vacuum and room temperature condition with low manufacturing cost [20,21]. Since it does not require any chemical etching and heat treatment, highly sensitive vibration sensors were achieved using porous properties of a printed pattern. Furthermore, the machining process can be directly monitored by simple attachment to the workpiece with desired location and direction. Since the sensor was attached to the workpiece in the form of a small thin film, the effect on the machining process and machine was also minimised. Furthermore, due to its flexible properties, the sensor can be attached on the free-form surfaces or hard-to-reach areas.

To validate the vibration measurement performance of a fabricated sensor, we measured relative resistance change of an AFN-printed pattern covered by ultraviolet (UV) resin. We demonstrated that the fabricated sensor was available to measure the vibration, according to the amplitude and frequency when using a vibration shaker. Then we applied the fabricated sensors to milling process monitoring using the most frequently used process parameters. In comparison to the commercial vibration sensor, the vibration data of a similar level was obtained with low fabrication and operation cost. Moreover, it was confirmed that the measured data varied according to various milling process parameters. Lastly, we applied a developed sensor to real-time quality monitoring of the turbine blade milling process.

## 2. Fabrication of a Low-Cost Vibration Sensor

### 2.1. AFN Printing System

Figure 1 shows the configuration of the AFN printing system, which consists of a nanomaterials feeder, nozzle, and vacuum chamber. Following successive repetition of excitation and purging, nanomaterials aerosolised by aerodynamic shock. Then, aerosolised nanomaterials move along the air flow generated by a pressure difference between the nanomaterials’ feeder and vacuum chamber, which remained at 1000 Pa and 400 Pa, respectively. Since it is a shock-induced solvent-free process, it is significantly important to maintain not only the pressure of each component but also the excitation and purging time at a specific level. In this study, we maintained excitation time and purging time for 10 ms and 90 ms, respectively. After the transportation of aerosol to the nozzle, aerosolised nanomaterials spurt from the nozzle and are aerodynamically-focused onto the substrate. The focused nanomaterials collide with the substrate at a high speed and are mechanically deposited onto a flexible substrate, which enables direct printing of various inorganic nanomaterials without chemical etching or heat treatment. Using mechanical translation of substrate with a feed rate of 0.2 mm$s^{-1}$ driven by multi-axis stage (SGSP20, Sigma Koki, Tokyo, Japan) at 1 mm below the fixed nozzle with an inner diameter of 150 μm (Taeha Co., Namyangju, Korea), a pattern with a line width of several tens of μm was printed. All systems were controlled by LabVIEW 2015 and NI USB 6009 modules (National Instrument, Austin, TX, USA). Since it is a direct printing technique for microscale patterns fabrication, the pattern with desired geometry can be printed at a desired position with a high degree of freedom in design. Moreover, it is possible to repair or reconfigure when it is necessary.

### 2.2. Vibration Sensor Fabrication

Figure 2a shows the schematic diagram of the fabrication process of an AFN-printed vibration sensor. In this case, we used silver nanoparticles (AgNPs) (<100 nm particle size, 576832, Sigma-Aldrich, St. Louis, MO, USA) as printing materials. To measure and monitor workpiece vibration by simple attachment with high thermal durability and electrical insulation, adhesive Kapton polyimide film (DuPont, Wilmington, DE, USA) was used as a flexible substrate. Then, the AFN printing process of AgNPs was followed on a desired position of substrate to fabricate the vibration sensor. Both ends of the vibration sensor were soldered using silver paste (conductive paste, 735825, Sigma-Aldrich, USA) and connected to an external data acquisition device by an electrical wire. After soldering and wiring for measurement, the printed sensor was covered by UV curable adhesive (UV-3300, Skycares Co., Gimpo, Korea) for electrical insulation and mechanical protection from chip and pneumatic pressure during the milling process. It was estimated that the production cost of the sensor including all processes was to be several cents per unit.

Figure 3a,b show the photograph of a fabricated vibration sensor obtained by an optical microscope (BX53M, Olympus, Tokyo, Japan) and a confocal microscope (OLS4100, Olympus, Japan), respectively. Since the entire length of the sensor printed on an adhesive tape was about 1 cm, it can be easily attached to the place to measure. Moreover, it can be attached to a certain level of free-form surface with a curvature using its flexible properties. Since the AFN printing process was conducted in a room temperature condition, it has its advantages of porous structure fabrication, as shown in Figure 3c,d. Using a porous structure composed of AgNPs, we can obtain highly sensitive properties of the printed sensor. In general, a metal NPs-based resistive type sensor has high sensitivity in comparison to a conventional sensor composed of metal foil [22]. This was due to a drastic change of contact resistance by mechanical detachment among NPs generated by mechanical elongation, as shown in Figure 3e. It was explained by an electron tunneling effect between adjacent NPs that occurred when their inter-particle gaps were shrunk. According to the quantum tunneling theory, the electrical resistance of metallic NP arrays can be denoted as shown in Equation (1) where ε is the distance between nanoparticles and β is the electron coupling term determined by the material properties, size of NPs, and ambient temperature [23].
(1)$$\frac{\Delta R}{R}=e^{\beta \Delta \varepsilon }-1$$

Since the electrical resistance is sensitive to the crevice size in electron tunneling, it was well explained that the NPs-based sensor exhibits high sensitivity and fast dynamic response to external stimulus [24]. Hence, the NPs-based sensor has its advantages of an ultrafast response of electrical resistance to external stimulus with high frequency without hysteresis that can be applied to reliable detection of vibrations as high as several tens of thousands kHz due to its sensing mechanism.

According to previous research, it was demonstrated that not only pattern geometry but also the electrical resistance could be controlled by regulation of the process parameter [25]. The process parameter used for this study is scan velocity, which is defined as a printed pattern length by process time. Since the high scan velocity occurs due to the highly sensitive properties of the sensor, we could achieve initial resistivity of 90.63 × 10^−8^ Ωm for the printed sensor by optimising the process parameter.

### 2.3. Validation of Vibration Measurement Performance

Using the vibration shaker (Vibration testing shaker, TIRA GmbH, Schalkau, Germany), the vibration measurement performance of the sensor was evaluated in various vibration frequencies and amplitudes. We attached the sensor directly to a vibration shaker and then applied various types of vibration continuously. Since it was a resistive type passive sensor with different initial resistance according to process parameters, we measured the relative resistance change, which is defined by subtracting the initial resistance value from the current resistance value divided by the initial resistance. First, we applied a vibration with the same amplitude of 8 μm at 100, 200, and 400 Hz, respectively. As a result of frequency analysis by fast Fourier transform (FFT), it was confirmed that the amplitude spectrum of a similar magnitude was obtained at each vibration frequency, as shown in Figure 4. Likewise, we demonstrated that amplitude spectrum was proportional to the actual vibration amplitude in the experiment with oscillations of 4, 8, and 12 μm at 200 Hz, as shown in Figure 5. Hence, we showed that the printed vibration sensor can measure vibrations in several hundreds of Hz with several μm amplitudes.

## 3. Milling Process Monitoring

### 3.1. Workpiece Attached Vibration Measurement

Figure 6 shows the experimental setup for a workpiece vibration measurement. The experiments were performed in a 3-axis milling machine (DMG MORI HSC 55 Linear, DMG MORI AKTIENGESELLSCHAFT, Bielefeld, Germany). A machine tool with an 8-mm diametre and four flutes (Pro Steel VHM-Schruppfräser HPC TiAlN, 203052, Hoffmann Group, Munich, Germany) and a workpiece of 200 × 150 × 100 mm^3^ in size (1.2738, 40CrMnNiMo8-6-4, Dörrenberg Edelstahl GmbH, Engelskirchen, Germany) were used. The printed sensors were directly attached to the workpiece in x, y, and z-directions without interruption to the milling process. To validate the performance of a printed sensor, a commercial three axes vibration sensor (Accelerometres, 356A26, PCB piezotronics, Depew, NY, USA) was attached next to the printed sensor. The data from the printed sensor and commercial vibration sensor was measured at a rate of 1,600 Hz per channel by NI USB 9162 carrier, NI 9234 modules, and DIAdem software (National Instruments, Austin, TX, USA).

We compared the performance of the directly printed sensor to a commercial vibration sensor in the x, y, and *z*-axis direction during the milling process, respectively, as shown in Figure 7. As shown in Figure 7a,c, the relative resistance change of the directly printed sensor exhibits a similar signal to the vibration measured by a commercial vibration sensor in the x-axis and y-axis directions. The vibration intensity was much greater in the milling state than in the idle state. However, since the milling process was only performed on a two-dimensional plane, there was less of a vibration measurement in the *z*-axis direction, as shown in Figure 7e. Comparing the measurement data of the commercial vibration sensor and directly printed sensor, it was confirmed that the power spectrum has peaks almost at an identical frequency regardless of the axis direction, according to the frequency power spectrum analysis. It was assumed that peaks of power spectrum between 400 Hz and 450 Hz are due to contact between the workpiece and the machine tool during the milling process. Hence, we have shown that the directly printed sensor can obtain the comparable data to the commercial vibration sensor at a much lower cost with an easy operation method, which was suitable for milling process monitoring.

### 3.2. Process Parametrisation

Further experiments are applied at the same experimental setup to analyze the sensibility and validity of the directly printed sensor using different milling conditions, as listed in Table 1. We varied four process parameters include RPM, feed rate, cutting depth, and cutting side. The parameters are chosen for their commonly used values and 15% to 20% above and below. 

Figure 8 shows the power spectrum analysis for each process parameter comparing the commercial vibration sensor and directly printed sensor. The experimental results for both sensors show a similar power spectrum. First, since the contact frequency between each tool flute and the workpiece depends on the spindle RPM, the peak frequency increased as the RPM grew while the intensity of the power was maintained at a certain level. However, the three parameters except RPM affected the power of the peak frequency rather than the position of the peak frequency itself. The power at the peak increased as feed rate, cutting depth, and cutting side increased. It was due to the increase of cutting force applied to the workpiece as each process parameter grows. The higher cutting force occurs as more strain on the workpiece, which leads to an increase of the vibration magnitude.

## 4. Application to Digital Twins

The main objective of all processes to manufacture high technology products complies with the specified ranges of permissible variation. Hence, there have been several efforts to construct the digital twin to measure the status of the product along the process and supply chains [26,27]. Since vibration plays an important role in turbine components machining due to its thin wall structures and high requirements of accuracy, the importance of the digital twin for turbine components machining is much higher than the other process.

Figure 9a,b shows experimental setup for monitoring of turbine blade milling process using directly printed sensors attached on the workpiece. In order to serve the mass production of turbine components and provide evidence for compliance with certification requirements, vibration data was obtained in real time during the entire milling process. A tool with a 12-mm diameter and 1.5-mm corner radius (GARANT VHM—Torusfräser, TiAlN, 206040, Hoffmann Group, Munich, Germany) and workpiece with 62 × 50 × 15 mm^3^ (EN AW 5083, AlMg4.5Mn0.7, GLEICH Aluminum GmbH, Kaltenkirchen, Germany) were used. The actual turbine blade milling process was conducted in order of roughing, pre-finishing, and finishing, as shown in Figure 9c–e. However, the data was analyzed only for the finishing process, which majorly affects the workpiece accuracy and surface quality.

As shown in Figure 10a,c, the milling process was performed multiple times with the same workpiece and machine tool with good surface quality and poor surface quality, respectively, by adjusting the process parameters. Since the bigger cutting depth was used for the machining of the turbine blade with poor surface quality, surface defects occurred while the overall shape was similar to the turbine blade with good surface quality. However, it is very important to maintain a good surface integrity where small surface defects can damage the overall turbo machinery system. The vibration data was obtained in real time by the directly printed sensor to monitor the vibrations during the machining process. We plotted the maximum value of the relative resistance change during the execution of each NC program line, as shown in Figure 10b,d. According to the results, much larger vibrations partly occurred at the turbine blade milling with poor surface quality in comparison to those of good surface quality. Moreover, it was confirmed that a larger vibration occurred when machining the upper part of the turbine blade like a general thin wall milling process. In addition, we could predict the actual visible surface defect using obtained data by the directly printed sensor, which shows that it can be applied as digital twins for various machining applications in the future.

## 5. Conclusions

Following the trends to the smart and intelligent manufacturing technology, the importance of process monitoring is increasing in various engineering fields. Among them, the milling process is one of the most widely used machining process and is expected as the most potential field along the digitisation of manufacturing due to their high requirements in quality. Since the traditional methods to measure the forces and vibrations are difficult to use in terms of cost and an installation method, there have been several efforts to develop the low-cost and easy-to-use sensors for milling process monitoring. Herein, we developed a low-cost vibration sensor using the AFN printing system with a high degree of freedom in design and manufacturing. Since it has its advantages of microscale porous pattern printing onto a flexible substrate, it was possible to measure small vibrations by simple attachment to the workpiece using the inherent properties of NPs. First, we evaluated the vibration measurement performance of a directly printed sensor using a vibration shaker by measuring the resolution and accuracy, according to the vibration frequency and amplitude. Then, we compared a directly printed sensor to a commercial vibration sensor in the milling process by attaching both to the workpiece. The relative resistance change of a directly printed sensor exhibits a similar signal to actual vibration measured by a commercial vibration sensor with low manufacturing and operation cost. Furthermore, an experiment comparing the commercial vibration sensor was performed according to various process parameters such as RPM, feed rate, cutting depth, and cutting side. The peak frequency increases as RPM grows while power intensity remained. In terms of feed rate, cutting depth, and cutting side, the intensity of the vibration increases at the same frequency due to the growth of cutting force as each process parameter rises. Lastly, we applied a directly printed sensor to a turbine blade milling process that is highly susceptible to vibration. Using the vibration data obtained by an attached sensor on turbine blades with good and poor surface quality, the surface defect during the milling process was predicted in real time. Hence, the directly printed sensor can be applied to the digital twins for various machining processes with low manufacturing cost and easy-to-use manner for smart manufacturing applications.

## Figures and Tables

**Figure 1 materials-13-02920-f001:**
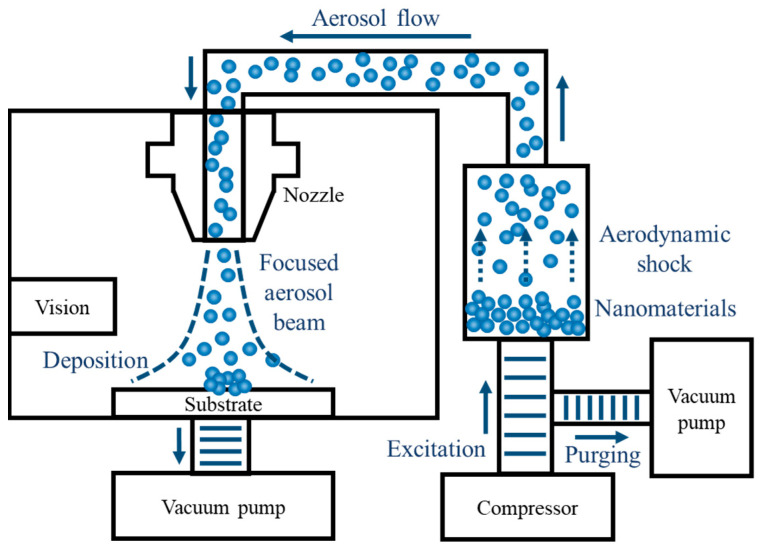
Schematic diagram of an aerodynamically-focused nanomaterials printing system.

**Figure 2 materials-13-02920-f002:**
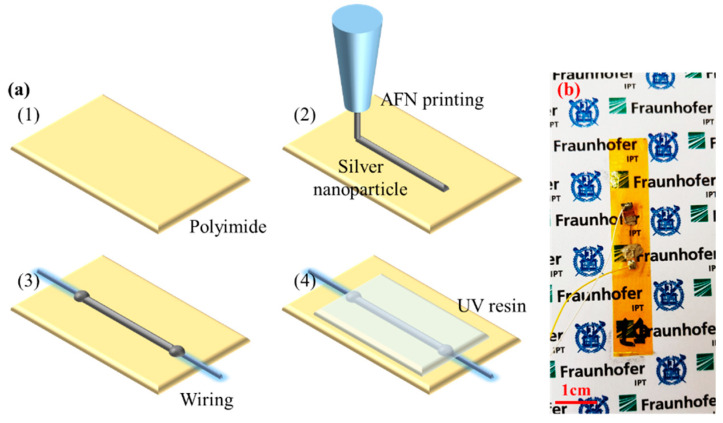
(**a**) The fabrication process of a directly printed vibration sensor. (**b**) The photograph of the fabricated vibration sensor.

**Figure 3 materials-13-02920-f003:**
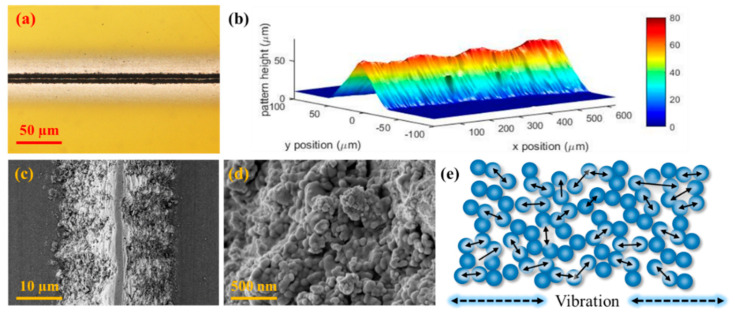
(**a**,**b**) The optical microscope image and confocal microscope image of the fabricated sensor, respectively. (**c**,**d**) The scanning electron microscopy (SEM) image and its magnified view of the fabricated sensor. (**e**) Schematic diagram of the sensing mechanism.

**Figure 4 materials-13-02920-f004:**
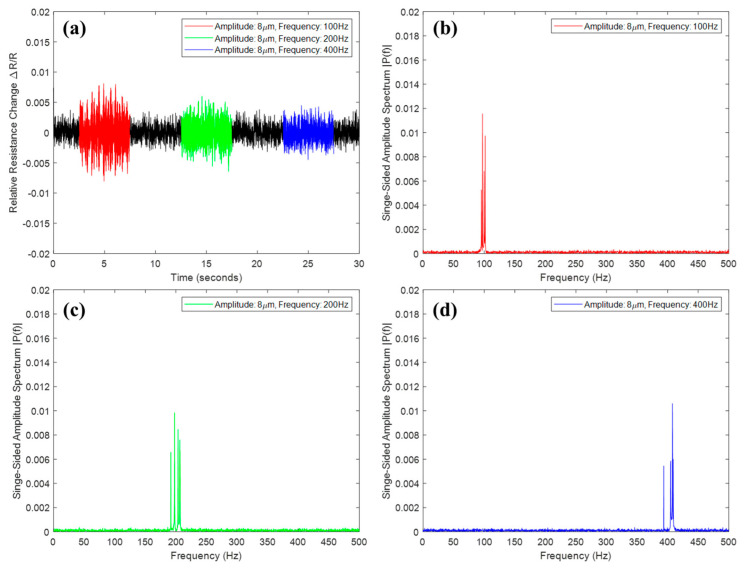
Experimental results according to the vibration frequency. (**a**) Relative resistance change, amplitude spectrum of (**b**) 100 Hz, (**c**) 200 Hz, and (**d**) 400 Hz.

**Figure 5 materials-13-02920-f005:**
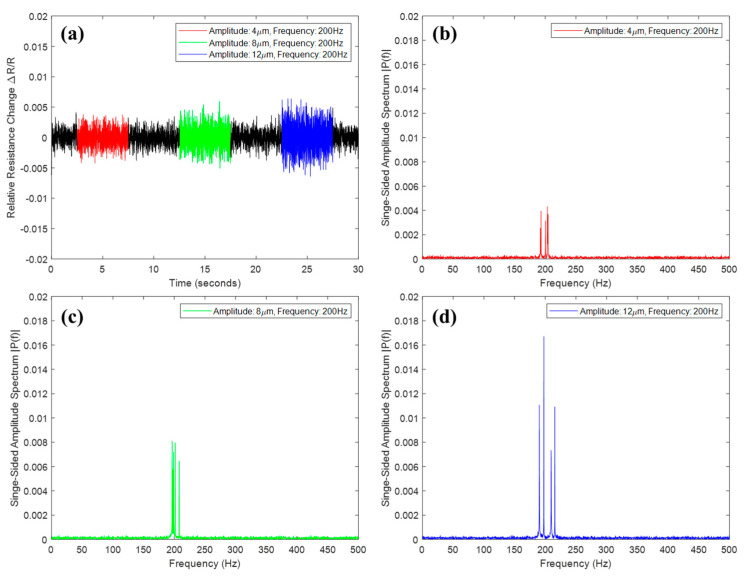
Experimental results according to the vibration amplitude. (**a**) Relative resistance change, amplitude spectrum of (**b**) 4 μm, (**c**) 8 μm, and (**d**) 12 μm.

**Figure 6 materials-13-02920-f006:**
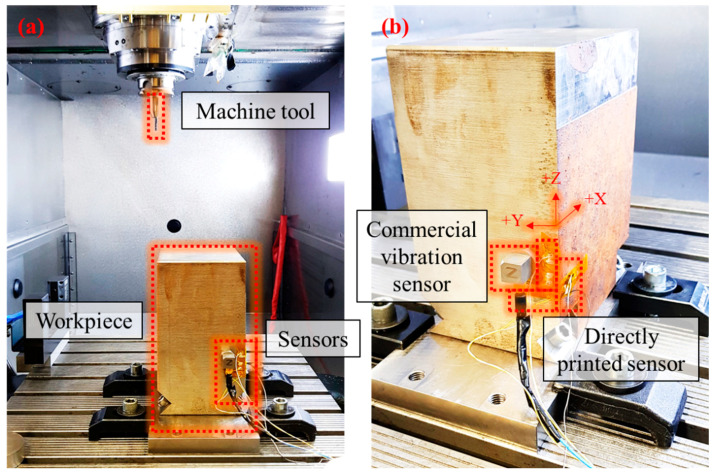
(**a**) Experimental setup for workpiece vibration measurement. (**b**) A magnified view of the attached commercial vibration sensor and directly printed sensors to the workpiece.

**Figure 7 materials-13-02920-f007:**
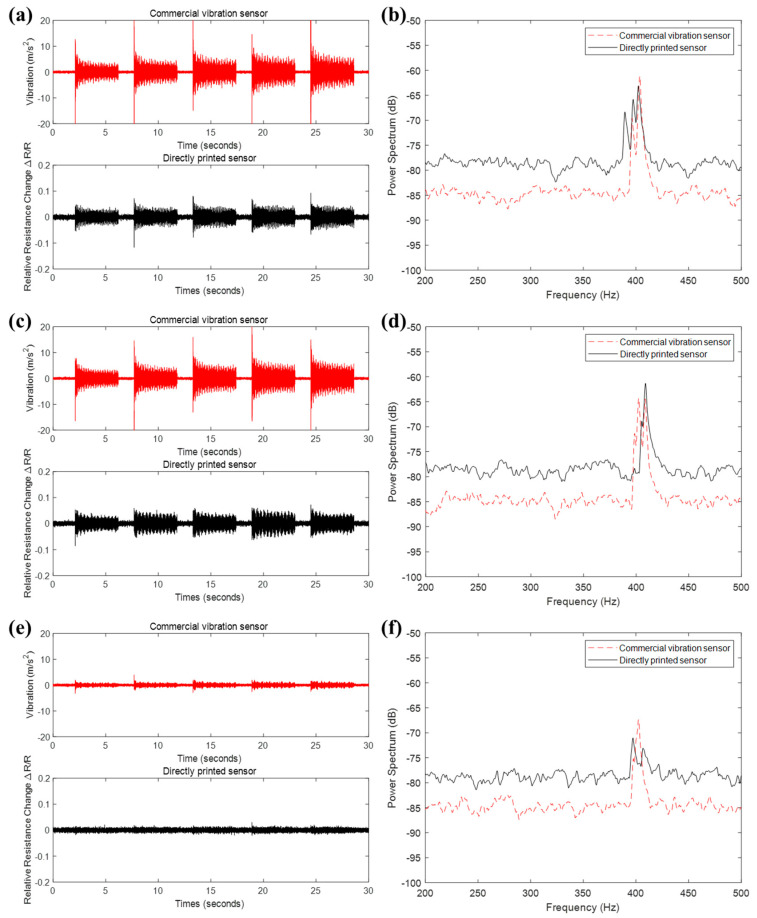
Comparison of experimental results of a commercial vibration sensor and directly printed sensor. (**a**,**b**) x-axis (**c**,**d**) y-axis, and (**e**,**f**) z-axis relative resistance change and power spectrum, respectively.

**Figure 8 materials-13-02920-f008:**
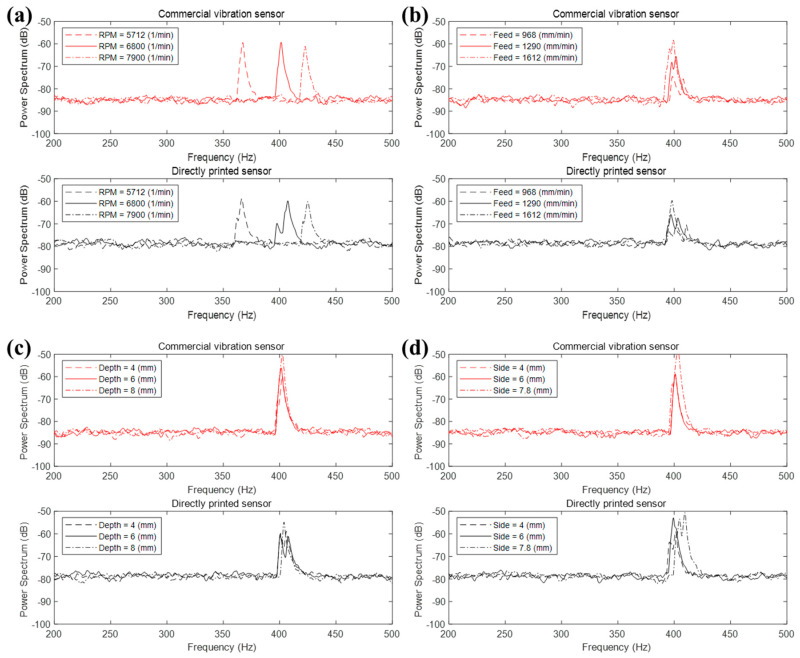
Power spectrum according to the process parameter. (**a**) RPM, (**b**) feed rate, (**c**) cutting depth, and (**d**) cutting side.

**Figure 9 materials-13-02920-f009:**
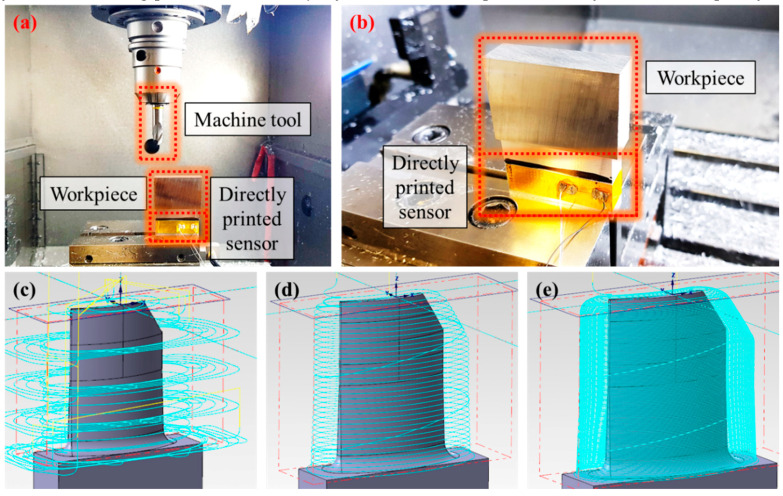
(**a**) Experimental setup for turbine blade milling. (**b**) A magnified view of the directly attached printed sensor to the workpiece. (**c**–**e**) A turbine blade milling procedure in sequence of roughing, pre-finishing, and finishing.

**Figure 10 materials-13-02920-f010:**
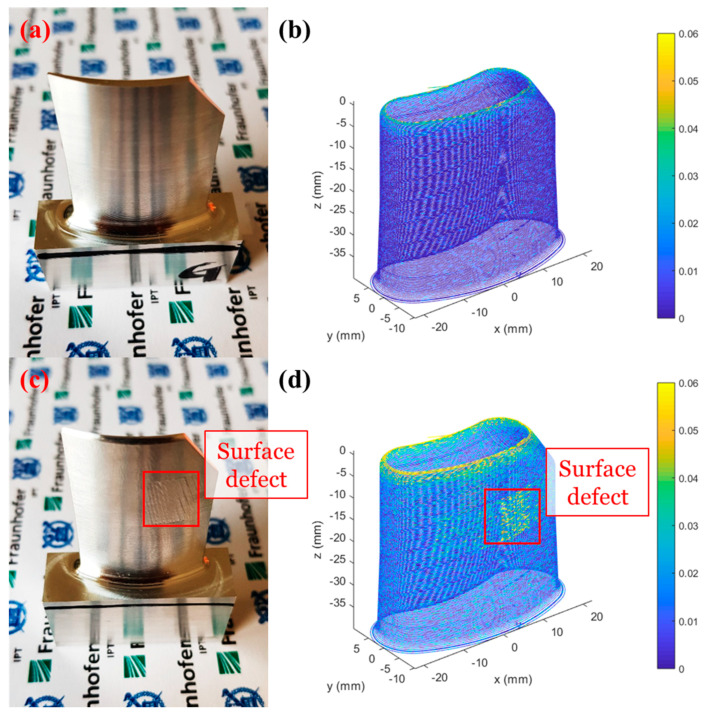
(**a**,**c**) The photograph of turbine blades with good and poor surface quality. (**b**,**d**) A real-time quality prediction based on vibration data measured by a directly printed sensor with good and bad surface quality.

**Table 1 materials-13-02920-t001:** Process parameter for milling process monitoring.

RPM (1/min)	Feed Rate (mm/min)	Cutting Depth (mm)	Cutting Side (mm)
5712	968	4	4
6800	1290	6	6
7900	1612	8	7.8

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
