# Peer review of "Directly Printed Low-Cost Nanoparticle Sensor for Vibration Measurement during Milling Process"

_materials, 2020, doi:10.3390/ma13132920_

Round 1
Reviewer 1 Report
The manuscript,Directly printed low-cost nanoparticle sensor for vibration measurement during milling process, presents a vibration sensor based on printing silver nanoparticles onto polyimide substrates with an aerodynamically focused nanomaterial printing system. The authors were able to demonstrate monitoring the vibration of a workpiece during the milling process. The manuscript can be considered for publication if the authors could address the following concerns properly.
First of all, the principle of the sensor is based on the resistance variation induced by the strain. What is the maximum vibration strength that could be detected without damaging the sensor? What is the thermal damage threshold of the sensor as the milling is typically accompanied by heating?
Second, a few important parameters of the sensors should be provided. For example, the thickness of the silver layer after printing, the width of the printed region, and the thickness of the substrate. Since the resistance is used to measure the vibration, the resistivity of the printed samples without vibration should be provided as a starting point in addition to the relative values.
Third, the authors should explain why the polyimide flexible film is chose as the substrate. For this specific application, vibration monitoring, flexible substrates may reduce the sensitivity. Besides, polyimide has a large thermal expansion coefficient, which might be detrimental.
Finally, the authors are encouraged to go through the manuscript again to correct grammar errors.
Author Response
Thank you for your comments concerning our manuscript entitled “Directly printed low-cost nanoparticle sensor for vibration measurement during milling process”. All of comments were valuable and helpful for revising and improving our manuscript as well as our future research. We have made corresponding modification or correction in the revised manuscript according to review comments, which were appended as following. Please consider the attachment.

Reviewer 2 Report
I read with interest the manuscript “Directly printed low-cost nanoparticle sensor for vibration measurement during milling process”. The authors demonstrate high-quality vibration sensors based on 3D printed Ag lines and show their performance on turbine blade manufacturing process.
The manuscript is of high quality and deserves to be published in its current state. Nevertheless, I would still recommend the authors to address following questions/remarks:
-> AFN printing system. Can the authors explain more in detail how the focusing of aerosol beam is performed? In addition, benchmarking of the technique to similar approaches (aerosol jet printing, for instance) would be of benefit.
-> Page 3, line 108. Can the authors break down the cost estimation per parts (processes, materials)?
-> Figure 3: Resolution of the printed line is rather impressive. How sensitive is the technique on the surface-state of the substrate? Did the authors pre-treat surface of Kapton?
-> Sensor performance. How did the authors attach the sensors on the objects of interests? Please add the supplier of the adhesive/glue and the procedure.
-> Can the authors report the absolute value of resistance of the sensor and resistivity of the printed line? Comparison with the bulk value would also be of interest.
Author Response

(The authors gave the same response as above.)

Round 2
Reviewer 1 Report
The authors have addressed all of my comments, and therefore, I would like to recommend this manuscript to be published.